# Using Skew to Assess the Quality of GAN-generated Image Features

**Lorenzo Luzi**\*                                                                                                    *enzo@rice.edu*
*Rice University*
*Pacific Northwest National Laboratory*

**Helen Jenne**\*                                                                                              *helen.jenne@pnnl.gov*
*Pacific Northwest National Laboratory*

**Carlos Ortiz Marrero**                                                                    *carlos.ortizmarrero@pnnl.gov*
*Pacific Northwest National Laboratory*
*North Carolina State University*

**Ryan Murray**                                                                                            *rwmurray@ncsu.edu*
*North Carolina State University*

**Reviewed on OpenReview:** *https://openreview.net/forum?id=Io3jDUC4DP*

## Abstract

The rapid advancement of Generative Adversarial Networks (GANs) necessitates the need to robustly evaluate these models. Among the established evaluation criteria, the Fréchet Inception Distance (FID) has been widely adopted due to its conceptual simplicity, fast computation time, and strong correlation with human perception. However, FID has inherent limitations, mainly stemming from its assumption that feature embeddings follow a Gaussian distribution, and therefore can be defined by their first two moments. As this does not hold in practice, in this paper we explore the importance of third-moments in image feature data and use this information to define a new measure, which we call the Skew Inception Distance (SID). We prove that SID is a pseudometric on probability distributions, show how it extends FID, and present a practical method for its computation. Our numerical experiments support that SID either tracks with FID or, in some cases, aligns more closely with human perception when evaluating image features of ImageNet data. Our work also shows that principal component analysis can be used to speed up the computation time of both FID and SID. Although we focus on using SID on image features for GAN evaluation, SID is applicable much more generally, including for the evaluation of other generative models.

## 1 Introduction

The goal of generative modeling is to learn a data distribution that generates novel samples which are indistinguishable from training data. Generative Adversarial Networks (GANs) (Goodfellow et al., 2014)) for image generation are widely known generative models that have seen enormous progress in recent years. As these models continue to evolve, the need for robust and informative evaluation metrics is increasingly important to benchmark different models.

Evaluating the quality of artificially generated samples is a multifaceted challenge. Ideally, a generative model should produce samples that not only appear to be from the training dataset, but also incorporate its full range of variations. Quantifying this is far from straightforward, leading to an active research domain focused on GAN evaluation. However, progress in this area is noticeably slower than the rapid advancements in the development of GAN models. In fact, one of the pioneering metrics in GAN evaluation, the Fréchet Inception Distance (FID) (Heusel et al., 2017), continues to be a well-regarded tool for GAN evaluation.

---

\*These authors contributed equally to this work.

FID separately embeds the generated and training samples into a feature space via a convolutional neural network (CNN), with Inception-v3 being a common choice. These feature representations are then modeled as multivariate Gaussian distributions. The actual FID score is determined by measuring the distance between the two Gaussian distributions using the Fréchet distance (also known as the 2-Wasserstein distance). It has been well established that FID correlates well with human judgement, is sensitive to small changes in the distribution (e.g. blurring or adding small artifacts to images), can detect intraclass mode collapse, and is fast to compute (Heusel et al., 2017; Lucic et al., 2018).

That being said, FID is not without shortcomings. Common critiques of FID include its high bias (typically requiring at least 50k samples for its computation), its sensitivity to the choice of CNN for feature extraction, and its presentation as a single numerical value without distinguishing between model fidelity and diversity. Over the years, other metrics have been proposed with their own trade-offs, such as Kernel Inception Distance (Sajjadi et al., 2018), Intrinsic Multi-Scale Distance (IMD) (Tsitsulin et al., 2019), Class-Aware Latent Distance (Swisher, 2022), and precision/recall-type metrics (Sajjadi et al., 2018; Kynkäänniemi et al., 2019; Naeem et al., 2020). For a more detailed discussion of related work, see Appendix A.

Our investigation is primarily concerned with the assumption of Gaussian characteristics in the embedded features and FID's failure to account for higher moments of these distributions. It is known that Inception-v3 features, among other classifier features, are skewed (Luzi et al., 2023). In this work, we explore the importance of third moment data for evaluating GAN performance. We introduce a novel measure, Skew Inception Distance (SID), which builds upon the foundation of FID but extends its scope to incorporate third-moment data, effectively accounting for skewness.

## 1.1 Overview and summary of contributions

We prove theoretical properties of SID in Section 2, in particular describing its properties as a metric. We review the relationship between probability distributions and their moments in Section 2.1, in order to define a metric on the space of moments and obtain a (pseudo)metric on the space of distributions. In Section 3, we provide a way to compute skewness by reducing the dimension of the Inception-v3 embeddings using principal component analysis (PCA) and provide two experiments showing that reducing the dimension of the embeddings does not lose important information. This result may be of independent interest in other problems that often rely on CNN embeddings, such as few-shot learning and out-of-distribution detection (Hu et al., 2021; Lee et al., 2018). Finally, in Section 4 we present empirical evaluations of SID, showing that it either behaves similarly to FID or more closely aligns with human perception.

## 2 Extending FID to include skewness

To compute FID, the set of real images and the set of generated images are featurized using the penultimate layer of Inception-v3, and the means $\mu_1$ and $\mu_2$ and covariance matrices $\Sigma_1$ and $\Sigma_2$ are estimated assuming that these features are Gaussian. Then FID is given by[1]

$$||\mu_1 - \mu_2||_2^2 + \text{Tr}\left(\Sigma_1 + \Sigma_2 - 2\sqrt{\Sigma_1\Sigma_2}\right). \tag{2}$$

In this paper, we seek to add a third term to FID to account for skewness. The challenges of this are first that we want to add a skewness term to Equation (2) in such a way that the resulting measure is a metric and second that the scale of the skewness term makes sense relative to the other terms.

In order to address these challenges we compute the coskewness tensor, $s \in \mathbb{R}^{n \times n \times n}$, defined by

$$s_{i,j,k} = \mathbb{E}[X_i^* X_j^* X_k^*]$$

---

[1]Although Equation (2) is the typical definition for FID (see e.g. Heusel et al. (2017)), we note that Fréchet distance between two multivariate Gaussian distributions $\mathcal{N}(\mu_1, \Sigma_1)$ and $\mathcal{N}(\mu_2, \Sigma_2)$ is defined as

$$||\mu_1 - \mu_2||_2^2 + \text{Tr}\left(\Sigma_1 + \Sigma_2 - 2\sqrt{\Sigma_1^{1/2}\Sigma_2\Sigma_1^{1/2}}\right) \tag{1}$$

The equivalence of Equations (2) and (1) is obvious when $\Sigma_1$ and $\Sigma_2$ commute. The equations are equivalent even when $\Sigma_1$ and $\Sigma_2$ do not commute, due to the fact that the eigenvalues of $\Sigma_1^{1/2}\Sigma_2\Sigma_1^{1/2}$ and $\Sigma_1\Sigma_2$ are identical. (see Wu & Koelzer (2022, Section 2.1.2) for further details).

where $X^* = (X_1^*, \ldots, X_n^*)$ is defined by $X^* = \Sigma^{-1/2}(X - \mu)$ and we compare the coskewness tensors of the sets of featurized real and generated images using the Frobenius norm of their element-wise cube root.

In order to prove that the resulting equation defines a pseudometric, we start by defining a metric on the space of moments of a distribution, which as we will see, allows us to obtain a pseudometric on the space of distributions. First, we need to formalize the relationship between probability distributions and their moments. The mathematical analysis of this relationship has been classically called the moment problem (Shohat & Tamarkin, 1950).

## 2.1 Theoretical set-up

The relationship between probability distributions and their moments is often utilized because metrics in families of probability distributions are more complex compared to metrics on subsets of $\mathbb{R}^n$. Even popular metrics, such as the Wasserstein distance, can be challenging to compute for many families of distributions. Consequently, tractable metrics on probability distributions start with a metric on the original probability space and reduce it to a metric on the moments or parameters of the distributions, which typically have closed-form solutions (e.g., Hellinger (Gibbs & Su, 2002), Bhattacharyya (Bhattacharyya, 1946), and Wasserstein (Villani, 2021) distances).

We provide several examples of distributions that can be identified with their moments, i.e., given a family of distributions $\mathbb{P}_\alpha$ indexed by $\alpha$, we can construct a bijection $f$ that maps $\mathbb{P}_\alpha$ to its moments $\mathbb{M}_\alpha$. Furthermore, this mapping is often a homeomorphism (meaning that both $f$ and its inverse are continuous) with the appropriate choice of metrics. When $f$ is a homeomorphism, we can inherit the topological properties of $\mathbb{M}_\alpha$ (Munkres, 2000). This means that in many ways, we can work in the space of the moments $\mathbb{M}_\alpha$, which is typically a subset of $\mathbb{R}^n$, instead of directly in the distribution space $\mathbb{P}_\alpha$. Not only that, but if the relationship is a homeomorphism, then we reduce the infinite-dimensional distribution space into a finite-dimensional vector space which has tractable methods for computing metrics. Below are some examples. The proofs of the claims made, including the explicit constructions of the homeomorphisms, are left to Appendix B.

**Example 1** (The univariate exponential distribution). The univariate exponential distribution is characterized by the following definition (Casella & Berger, 2021):

$$p_\lambda(x) = \lambda \exp(-\lambda x) \quad \text{for} \quad \lambda \in (0, \infty). \tag{3}$$

There exists a bijection from the distributions $\mathbb{P} = \{p_\lambda : \lambda \in (0, \infty)\}$ to the parameter space $(0, \infty)$, making exponential distributions completely characterized by their mean $\frac{1}{\lambda}$. Moreover, this bijection is a homeomorphism between $((0, \infty), |\cdot|)$ and $(\mathbb{P}, \|\cdot\|_{L^2(\mathbb{R}_{>0})})$.

**Example 2** (The multivariate Gaussian). The multivariate Gaussian distribution is characterized by the following definition (Mardia et al., 1979):

$$p_{\boldsymbol{\mu}, \boldsymbol{\Sigma}}(\boldsymbol{x}) = |2\pi\boldsymbol{\Sigma}|^{-\frac{1}{2}} \exp\left(-\frac{1}{2}(\boldsymbol{x} - \boldsymbol{\mu})^\top \boldsymbol{\Sigma}^{-1}(\boldsymbol{x} - \boldsymbol{\mu})\right) \tag{4}$$

for mean vector $\boldsymbol{\mu} \in \mathbb{R}^p$ and covariance matrix $\boldsymbol{\Sigma}$ in the space of $p \times p$ real, positive-definite matrices $\mathbf{S}_{++}^p$. The function $f$ that maps $p_{\boldsymbol{\mu}, \boldsymbol{\Sigma}}$ to $(\boldsymbol{\mu}, \boldsymbol{\Sigma})$ is an isometry between the space of multivariate Gaussian distributions equipped with the 2-Wasserstein metric and the space $(\boldsymbol{\mu}, \boldsymbol{\Sigma})$ equipped with the metric defined by

$$d\left((\boldsymbol{\mu}_1, \boldsymbol{\Sigma}_1), (\boldsymbol{\mu}_2, \boldsymbol{\Sigma}_2)\right) = \|\boldsymbol{\mu}_1 - \boldsymbol{\mu}_2\|_2^2 + \operatorname{Tr}\left(\Sigma_1 + \Sigma_2 - 2\sqrt{\Sigma_1 \Sigma_2}\right).$$

It is important to note that when comparing the moments of two distributions, they typically come from the same family of distributions. Let us consider two homeomorphisms: a homeomorphism $f$ from the set of all normal distributions $\mathbb{P}_n$ in $\mathbb{R}$ to its moments $\mathbb{R} \times (0, \infty)$ and a homeomorphism $f'$ from the set of all uniform distributions $\mathbb{P}_u$ in $\mathbb{R}$ to its moments $\mathbb{R} \times (0, \infty)$. If we have $p \in \mathbb{P}_n$ and $q \in \mathbb{P}_u$, we must have that $p \neq q$ because they are different distributions; however, we may have that their moments match exactly. This is because $f$ and $f'$ are different homeomorphisms operating on different spaces, and thus we must take care to consider not only the matching of moments but also the underlying distributions they represent.

## 2.2 Defining SFD: A metric on the first three moments of probability distributions

Let $\mathbb{P}$ denote a set of probability measures $\mathbb{P}$ that are homeomorphic to a space $\mathbb{M}$ equipped with a metric $d_{\mathbb{M}}$, which is the space of moments (or parameters) of the distribution. Denoting the homeomorphism as $f : \mathbb{P} \to \mathbb{M}$, we can define a metric by

$$d_{\mathbb{P}}(q, p) \triangleq d_{\mathbb{M}}(f(q), f(p)). \tag{5}$$

Essentially, we use $f$ to push the points $p, q$ to the moment space and then apply our metric there. Most importantly, calculating $f(q)$ and $f(p)$ is very easy and tractable; we simply calculate the moments in the usual statistical way, for example, using maximum likelihood estimation (Casella & Berger, 2021).

**Proposition 1.** *Equation* (5) *defines a metric* $d_{\mathbb{P}}$.

*Proof.* It is immediate that $d_{\mathbb{P}}$ is non-negative, symmetric, and satisfies the triangle inequality. The fact that $d_{\mathbb{P}}$ is definite follows from the injectivity of $f$. $\qquad\square$

As is clear from the proof, rather than requiring a homeomorphism one could consider an injective $f$ instead at the cost of losing the topological equivalence of $\mathbb{P}$ and $\mathbb{M}$. However, this is not necessary in many cases as a desired homeomorphism exists.

In the case of numerous distributions, a complete description requires multiple moments, so we view our space $\mathbb{M}$ as a product of several spaces: $\mathbb{M} = \mathbb{M}_1 \times \ldots \times \mathbb{M}_m$. Moreover, each space $\mathbb{M}_i$ may have its own metric $d_{\mathbb{M}_i}$. Therefore, we will need to consider which metric product space we will use. For any $p \in [1, \infty)$, we can define a product metric (Mendelson, 1990) as

$$d_{\mathbb{M}}\Big((\boldsymbol{x}_1, \ldots, \boldsymbol{x}_m), (\boldsymbol{y}_1, \ldots, \boldsymbol{y}_n)\Big) = \sqrt[p]{\sum_{i=1}^{m} d_{\mathbb{M}_i}^p(\boldsymbol{x}_i, \boldsymbol{y}_i)}.$$

Since all norms on finite-dimensional vector spaces are equivalent (Royden & Fitzpatrick, 1988), we will just pick $p = 2$. Therefore, given each moment space $\mathbb{M}_i$, we can construct a metric on that space $d_{\mathbb{M}_i}$ and together construct a metric $d_{\mathbb{M}}$ on the whole moment space $\mathbb{M}$ which we use to define a metric (Equation (5)) on the distributions themselves.

We are now ready to define our metric, which we call SFD (Skew Fréchet Distance), using the first three moments of a probability distribution:

$$d_{\mathbb{M}}^2((\boldsymbol{\mu}_1, \Sigma_1, \mathbf{S}_1), (\boldsymbol{\mu}_2, \Sigma_2, \mathbf{S}_2)) = \|\boldsymbol{\mu}_1 - \boldsymbol{\mu}_2\|_2^2 + \mathrm{Tr}\left(\Sigma_1 + \Sigma_2 - 2\sqrt{\Sigma_1\Sigma_2}\right) + \left\|\sqrt[\circ 3]{\mathbf{S}_1} - \sqrt[\circ 3]{\mathbf{S}_2}\right\|_F^2. \tag{6}$$

Here, $\sqrt[\circ 3]{\mathbf{S}}$ is the element-wise cube root[2] of the skew 3-tensor $\mathbf{S}$, which ensures that each term of SFD has the same units. (The coskewness 3-tensor $\mathbf{S}$ has cubed units, while the first and second terms in SFD have linear units before they are squared).

Note that SFD is a generalization of the 2-Wasserstein distance for Gaussians: since $\mathbf{S} = \mathbf{0}$ in the Gaussian case, SFD between two Gaussian is the same as the 2-Wasserstein distance.

The first and second terms of SFD define metrics on $\mathbb{R}^n$ and $\mathbf{S}_{++}^p$ (Givens & Shortt, 1984). So it remains to show that the third term defines a metric. We actually prove a more general result.

**Theorem 1.** *Let* $\mathbb{X}(n_1, \ldots, n_r; \mathbb{C})$ *be the set of all complex-valued, $r$-tensors with shape* $(n_1, \ldots, n_r)$. *Let* $\alpha : \mathbb{C} \to \mathbb{C}$ *be any invertible function. Then for* $\mathbf{A}, \mathbf{B} \in \mathbb{X}(n_1, \ldots, n_r; \mathbb{C})$ *we have that*

$$d(\mathbf{A}, \mathbf{B}) := \sqrt{\sum_{\boldsymbol{i} \in \{1, \ldots, n_1\} \times \ldots \times \{1, \ldots, n_r\}} \left|\alpha(\mathbf{A}_{\boldsymbol{i}}) - \alpha(\mathbf{B}_{\boldsymbol{i}})\right|^2}$$

*defines a metric* $d$.

---

[2]To our knowledge, there is no computationally efficient 3-tensor cube root.

We postpone the proof to Appendix C.

The metric $d$ looks very similar to a Frobenius norm, but it does not induce a norm. If we choose $\alpha$ to be the identity map, then we do indeed recover the Frobenius norm. However, there are many choices of $\alpha$ which make it so that $d$ does not induce a norm; for example, if $\alpha(x) = e^x$ we see that the homogeneity property of norms is violated.

For SFD, we choose $\alpha : \mathbb{R} \to \mathbb{R}$ defined by $\alpha(x) = \sqrt[3]{x}$ and see that SFD is a proper metric with the same units for each term. Additionally, we transform the skew term $s = \left\| \sqrt[\odot 3]{\mathbf{S}_1} - \sqrt[\odot 3]{\mathbf{S}_2} \right\|_F^2$ by $s' = \sigma\left(\frac{\alpha s}{m}\right) m - \frac{m}{2}$ using a sigmoid $\sigma$. SFD is still a metric since these are invertible transformations which preserve non-negativity. We do this to ensure that the skew term has comparable scaling to the covariance and mean terms. We saw that values of $\alpha = 10{,}000$ and $m = 150$ seemed to work well in the examples we tested.

If a distribution is homeomorphic to its first three moments, then SFD defines a metric on that family of distributions. However, if this is false and a distribution is homeomorphic to more than three moments, then SFD instead defines a pseudometric. More formally,

**Proposition 2.** *Suppose $\mathbb{P}$ defines a family of distributions that are defined by their first three moments, i.e., for all $p_1, p_2 \in \mathbb{P}$, we have that $(\boldsymbol{\mu}_1, \boldsymbol{\Sigma}_1, \mathbf{S}_1) = (\boldsymbol{\mu}_2, \boldsymbol{\Sigma}_2, \mathbf{S}_2)$ if and only if $p_1 = p_2$. Then, SFD defines a proper metric. If there are additional moments which are needed for the equality to hold, then SFD defines a pseudometric.*

*Proof.* In the case where $\mathbb{P}$ defines a family of distributions which are defined by their first three moments, SFD defines a metric because it is a product metric (Royden & Fitzpatrick, 1988), and by Theorem 1, each term in the product is a metric.

Now suppose $p_1$ and $p_2$ are defined by their moments $(\boldsymbol{\mu}, \boldsymbol{\Sigma}, \mathbf{S}, \mathbf{X}_1)$ and $(\boldsymbol{\mu}, \boldsymbol{\Sigma}, \mathbf{S}, \mathbf{X}_2)$; all their moments are the same except for $\mathbf{X}_1$ and $\mathbf{X}_2$ (which can be any order), and $\mathbf{X}_1 \neq \mathbf{X}_2$. Since SFD is zero in this case, SFD is not strictly definite and, therefore, SFD cannot be a metric. However, SFD defines a product over pseudometrics and therefore is a pseudometric itself (Freiwald, 2014). $\qquad\square$

The following example gives an instance where SFD is a pseudometric.

**Example 3.** Let $p_1$ and $p_2$ be univariate Gaussian mixtures with two components. The parameters of $p_1$ are $\mu_1 = \mu_2 = 0$, $\sigma_1 = \sigma_2 = 1$, and $\pi_1 = \pi_2 = 0.5$; that is the first GMM is a standard normal Gaussian. The parameters of $p_2$ are $\mu_1 = \mu_2 = 0$, $\sigma_1 = \frac{1}{\sqrt{2}}$, $\sigma_2 = \sqrt{2}$, $\pi_1 = \frac{2}{3}$, and $\pi_2 = \frac{1}{3}$. Both of these distributions have mean 0, variance 1, and skew of 0 meaning that SFD between these two distributions is zero even though they have non-matching higher-order moments.

While this is a limitation of SFD, it can still be used to compare distributions even when it is a pseudometric. For example, the Fréchet distance is commonly used on distributions that are not Gaussian.

Although our pseudometric SFD is quite general, for the remainder of the paper, we will apply SFD to Inception-v3 embeddings and refer to this specific case as SID.

## 3  Using dimensionality reduction to compute SID

To compute the coskewness tensor in practice, we need to reduce the dimension of the Inception-v3 embeddings, as these embeddings are 2048-dimensional. This requires the coskewness tensor to be $2048 \times 2048 \times 2048$, which is 64 GB using 32-bit precision. That is just to store one of these tensors, let alone store and manipulate two of them! Storing tensors of this size on the GPU is out of the question.

In addition, computing the coskewness tensor is the slowest of the three terms; the operation scales as $O(d^3)$, where $d$ is the dimensionality of the reduced feature space. In comparison, FID can be computed in $O(d^2 m + m^3)$ time, where $m << d$ (Mathiasen & Hvilshøj, 2020). (Although this requires observing that it suffices to compute the eigenvalues of $\Sigma_1 \Sigma_2$; many FID implementations compute $\text{Tr}(\sqrt{\Sigma_1 \Sigma_2})$ using the `scipy.linalg.sqrtm` function, which requires computing the Schur decomposition of $\Sigma_1 \Sigma_2$ and takes $O(d^3)$ time). Because of the memory and computational obstacles of SID, we reduce the dimension of the Inception-v3 embeddings.

**Table 1:** PCA reduces the dimension of the features while still maintaining important information. This table demonstrates the classification performance of Inception-v3 on ImageNet using features that have been reduced in dimension with PCA and then linearly projected back into their 2048-dimensional space. We see that the dimensionality reduction does not significantly affect classification performance, even for large reductions, implying that important information in the features is preserved.

| Dimensionality | Top-1 Train Acc | Top-5 Train Acc | Top-1 Val Acc | Top-5 Val Acc |
|---|---|---|---|---|
| 2048 (baseline) | 91.78% | 99.16% | 77.21% | 93.53% |
| 1024 (50%) | 90.77% | 98.95% | 76.24% | 92.83% |
| 512 (25%) | 88.48% | 98.21% | 74.15% | 90.82% |
| 256 (12.5%) | 81.59% | 95.12% | 68.40% | 85.55% |

In order to reduce the dimensionality of the Inception-v3 embeddings, we apply principal component analysis (PCA). PCA is a widely used dimensionality reduction method that projects the data onto a lower-dimensional subspace in such a way that maximizes the variance preserved. Because PCA maximizes the preserved variance, it is a natural choice for dimension reduction in our case (the intuition here is that, because FID is a function of the variances, if we assume that our variances will decay rapidly, truncating tail variances will not change FID by very much). After applying PCA, we need to normalize the covariance and skew terms. Because we center our data matrix, we do not modify the mean with the dimensionality reduction, just the covariance and skew.

Letting $X \in \mathbb{R}^{n \times p}$ denote our centered data matrix (where $n > p$), we will use the following transformation of $X$ to normalize the FID values for different values of $k$:

$$Y = \sqrt{\frac{\sum_{i=1}^{p}[D]_{ii}}{\sum_{i=1}^{k}[D_1]_{kk}}} X V_1. \tag{7}$$

Here, $V_1$ is the matrix consisting of the first $k$ right singular vectors of $X$, $D$ is the matrix of singular values, and $D_1$ is the $k \times k$ truncated matrix of singular values. We discuss this normalization further in Appendix D.

We emphasize that for a fixed $k$, we are transforming both sets of features in the same way so that they are comparable.

To summarize, to compute SID we first compute the Inception-v3 embeddings $X$, then compute the singular value decomposition of $X$ in order to apply the transformation in Equation (7). After the dimensionality of the embeddings has been reduced, we compute the FID terms and the coskewness tensor.

### 3.1 The effect of PCA on FID

Before using the reduced-dimension features to compute SID, we needed to ensure that PCA reduces the dimension of the features while still maintaining important information. We investigate this question in two ways. First, since Inception-v3 is a classifier, a natural question is how the classifier performance deteriorates as we reduce the feature dimension. To implement this experiment, we first calculate the PCA transform for $k \in \{2048, 1024, 512, 256\}$ on the training data. Then, we extracted Inception-v3 features from the training and validation datasets. Next, we reduced the dimension with the precalculated transform and then projected back to the 2048-dimensional space; since the PCA transform is a semi-orthogonal matrix $V_k$, we simply multiplied by the transpose $V_k^\top$. Finally, we sent these features to the final layer of Inception-v3 to calculate training and validation accuracies. The results are summarized in Table 1. We see that even with significant dimensionality reduction (12.5%), there is *at most* a $10\%$ difference in accuracy, which implies that using PCA to reduce the dimension to 256 does not significantly affect the classification power of the embeddings.

Furthermore, we investigated how dimension reduction affected the behavior of FID on common experiments used to compare GAN metrics. These experiments involve corrupting an image and studying how the GAN metric behaves with respect to a parameter controlling the level of corruption. Specifically, we used the following corruptions: Gaussian noise, salt and pepper noise, Gaussian blur, and adding rectangles at randomly chosen locations (Heusel et al., 2017). We describe these experiments in more detail in Section 4.2. In all experiments, the dimension reduction affected the value of FID slightly but not its behavior; examples are shown in Figure 1.

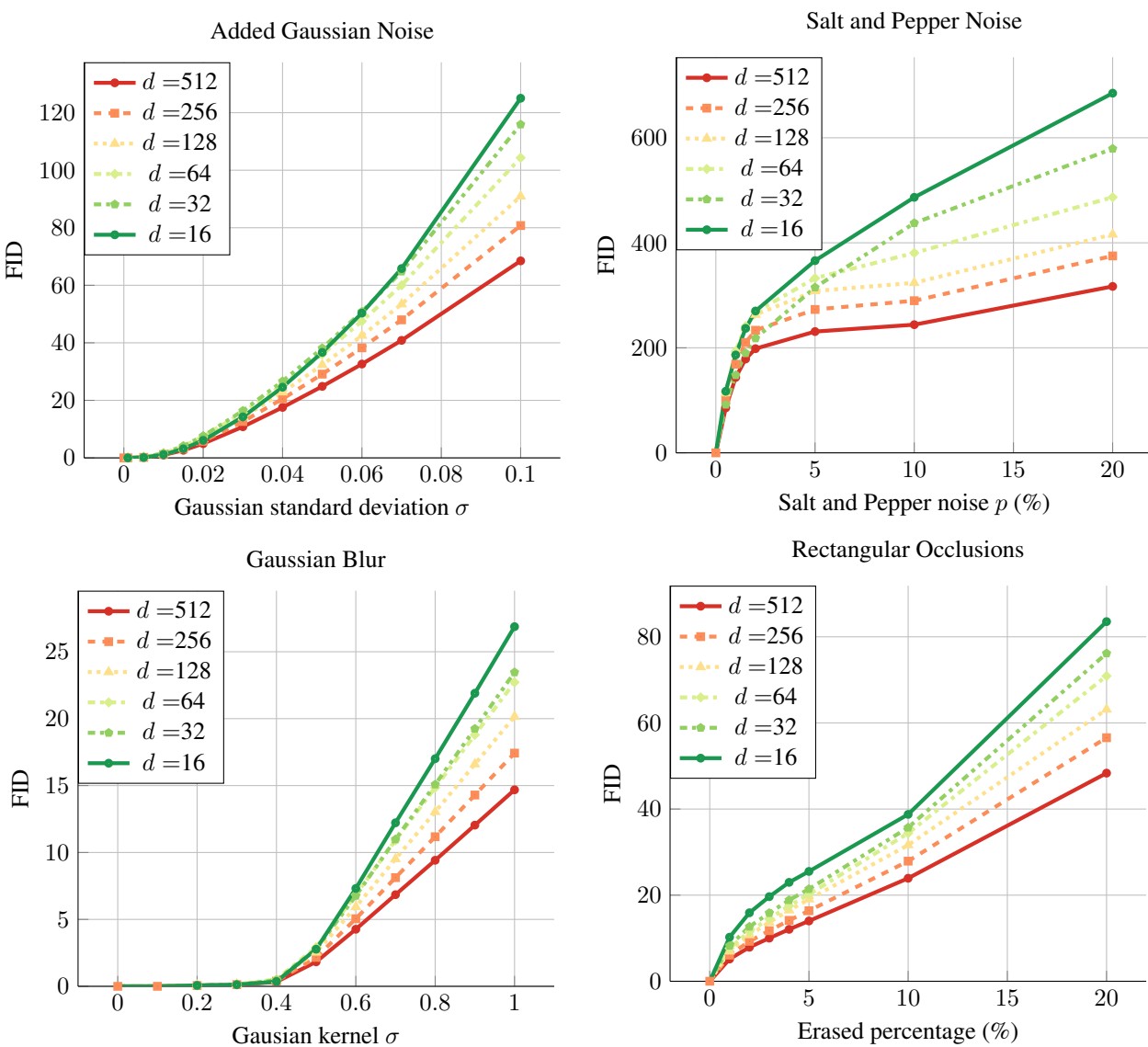

**Figure 1:** FID behaves similarly even when we project features down to lower dimensions via PCA. We show four types of distortions that are applied to ImageNet images: added Gaussian noise, salt and pepper noise, Gaussian blur, and rectangular occlusions. For the Gaussian blur, we use a kernel size of 5. In each plot, FID is shown as a function of the parameter controlling the distortion, after reducing the embeddings to dimension $d \in \{512, 256, 128, 64, 32, 16\}$. All experiments are done using Inception-v3 as a feature extractor on the entire (50K) ImageNet validation set.

## 3.2 Time improvement for SID

Table 2 demonstrates that dimensionality reduction leads to significant time and memory savings when computing skewness. These savings are so substantial that the skew calculation can be computed completely on a GPU.

Having established that using PCA to reduce the dimensionality of Inception-v3 features is reasonable and has computational and memory benefits, we turn our attention to the behavior of SID as a performance measure.

**Table 2:** Calculating skewness on the GPU is computationally efficient, especially when reducing the feature dimensionality. There is an order of magnitude difference in CPU and GPU times. ($*$) would not fit completely on the GPU but can be calculated on a GPU using mini-batches; however, batch and mini-batch algorithms are not comparable due to intrinsic differences.

| Dimensionality | | CPU time | GPU time | Memory |
|---|---|---|---|---|
| 2048 | (baseline) | 886.9s | $*$ | 64 GB |
| 1024 | (50%) | 161.1s | 9.1s | 8 GB |
| 512 | (25%) | 35.5s | 0.8s | 1 GB |
| 256 | (12.5%) | 4.66s | 0.02s | 128 MB |

## 4  SID Experiments

### 4.1  Features are skewed even after PCA

It is known that Inception-v3 features, among other classifier features, are skewed (Luzi et al., 2023). However, it is not immediately clear whether features are skewed after dimensionality reduction using PCA. In order to test this, we applied standard skewness tests to the dimension-reduced data. In particular, we performed the Mardia skewness test (Mardia, 1970) on Inception-v3 multivariate 2048-dimensional features with a significance level of $0.1\%$ and the data failed the test at dimensionality reductions of $1024, 512, 256, 128, 64, 32$, and $16$. We also performed Kolmogorov–Smirnov hypothesis tests[3] (Dodge, 2008) at each marginal with significance values of $0.1\%$ and found that $100\%$ of the marginals failed the tests. This demonstrates that even with extreme dimensionality reduction with PCA, skewness persists in the data. We repeated the experiment with ResNet-18 features and obtained the same result: all tests failed with significance values of $0.1\%$.

### 4.2  Comparing SID and FID

In order to compare SID and FID we studied how these measures are affected by the corruptions introduced in Section 3.1:

1. Gaussian noise: We added Gaussian noise with $\sigma \in \{0, 0.001, 0.005, 0.01, \ldots, 0.1\}$.
2. Salt and pepper noise: We changed a proportion $p$ of the pixels in the image to black or white (with equal probability) for $p \in \{0, 0.5\%, 1.0\%, 1.5\%, \ldots, 20\%\}$.
3. Gaussian blur: We convolved the image with a Gaussian kernel with standard deviation $\sigma \in \{0.1, 0.2, 0.3, \ldots, 1.0\}$.
4. Adding black rectangles as occlusions: We added five rectangles at randomly chosen locations, where the scale increases with scale parameter $s \in \{1\%, 2\%, 3\%, \ldots, 20\%\}$.

In these experiments, we found that the skew term (and consequently SID) often behaved similarly to FID.

In other work, larger values were chosen for the parameters controlling these distortions (see Heusel et al. (2017)). But looking at examples of images with these levels of corruption shows that for most values of these parameters, the difference between the corrupted image and the original image is easily perceptible by a human. This is why in order to investigate what is happening at the transition point between noise that is human perceptible, we focused on smaller values of these parameters. Here we see a difference between the skew term and the FID. For example, in Figure 2, which shows the FID and skew term as Gaussian noise is added with $\sigma \in \{0.01, 0.015, 0.02, 0.03, 0.04, 0.05, 0.06\}$, we see that the skew does not pick up any difference until the noise becomes perceivable. Once the noise becomes perceivable, the skew increases faster than the covariance and mean terms. The Gaussian blur experiment with $\sigma \in \{0.0, 0.1, 0.2, \ldots, 1.0\}$ showed similar results and is shown in Appendix E. Other experiments showed that SID and FID behaved similarly, as shown in Figure 4 in Appendix E.

While in this section, we focused on results for SID using Inception-v3 features on ImageNet data, we also studied SID using different models for feature extraction (namely, ResNet-18, ResNet-50, and ResNeXt-101 (32×8d)) and differ-

---

[3]The Kolmogorov-Smirnov hypothesis test is a nonparametric method for evaluating the goodness-of-fit. It assesses whether an underlying probability distribution (the marginal distributions in our case) differs from a hypothesized distribution, a Gaussian distribution.

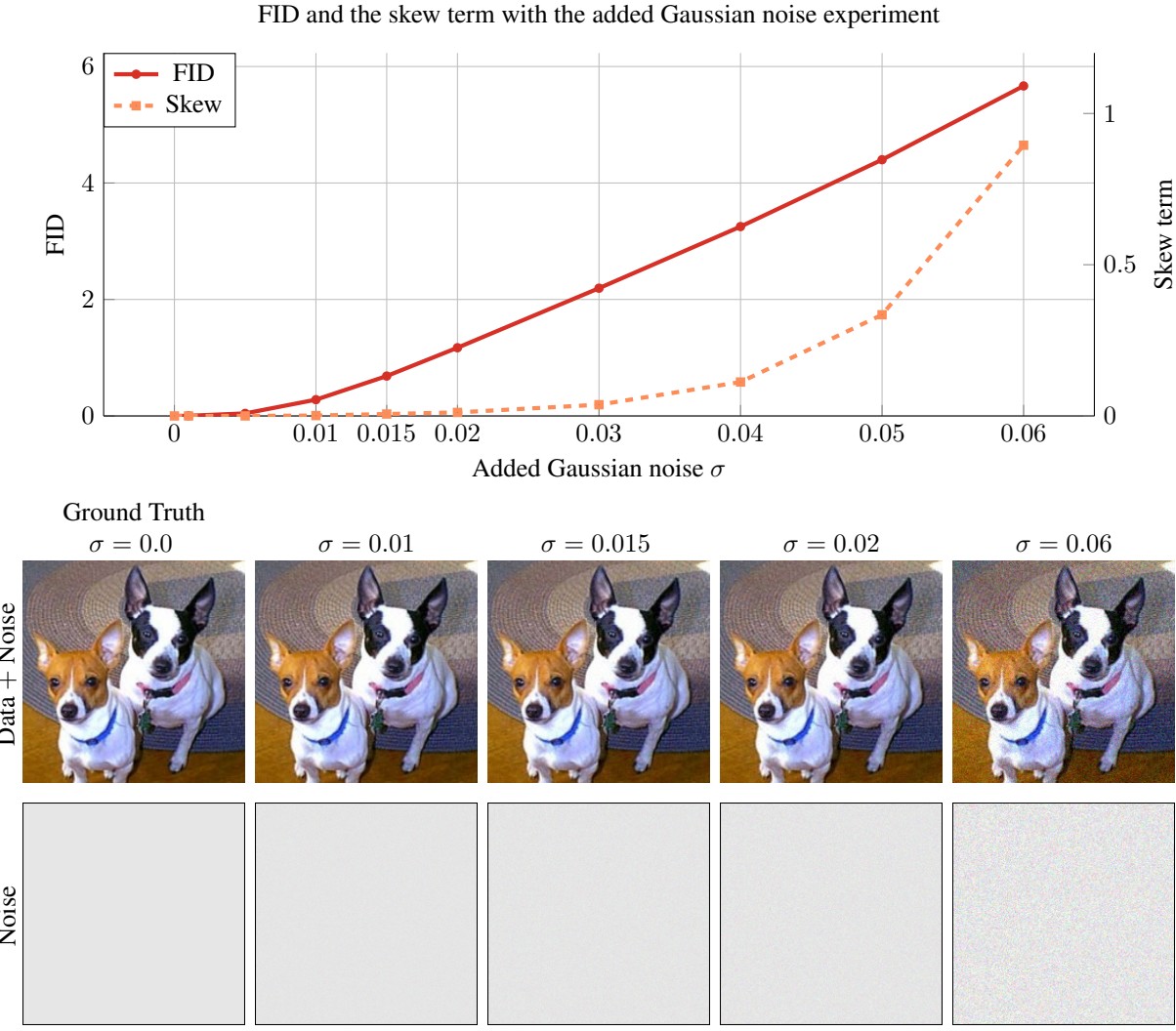

**Figure 2:** Skew tracks more with human perception than FID. The noise levels on the images are quite low and undetectable until $\sigma > 0.015$; yet the FID increases linearly throughout. On the other hand, skew stays low for these undetectable noise levels. These were typical images taken from the 50,000 samples used to calculate FID.

ent datasets (CelebA (Liu et al., 2015), FFHQ (Karras et al., 2019), and SVHN (Netzer et al., 2011)). Additionally, to explore SID's applicability to different types of generative approaches, we studied SID using data from diffusion models, using pretrained diffusion models that were trained on Stanford Cars, CelebA, AFHQ, and the Flowers dataset to generate images. These experiments, presented in Appendices F and G, show that SID is robust to the model used for feature extraction and that in all settings SID increases as the noise distortions increase.

## 5 Conclusion

This paper presents a novel metric on distributions of image features that can successfully incorporate third-moment data and can be used to evaluate the performance of GANs. Our findings indicate that, particularly when images are subtly distorted, SID can sometimes better match human perception. A notable limitation of SID is its computational complexity, which is $O(d^3)$, where $d$ is the dimensionality of the features, making dimensionality reduction essential for its calculation. Consequently, the dimensionality reduction we perform in estimating SID may not include information that is not captured in the largest principal components. Another limitation of our work can be observed in many related works: experiments often struggle to faithfully replicate the nuanced distinctions between real and

GAN-generated images; for instance, GAN images may exhibit multiple types of distortion in the same image. Furthermore, assertions regarding alignment with human perception should be substantiated through human studies. The field would greatly benefit from an extensive and standardized repository of experiments designed to evaluate GAN metrics. Until then, it is crucial that GAN evaluation metrics are based on a solid theoretical foundation.

The current work does not compare SID to other established metrics, which is an important direction for future study. We conclude by discussing three other directions for future work:

**Exploring other formulations of the skew term**
The subtle differences between SID and FID have raised the question of how much redundant information the skew term encapsulates. We're considering whether our current formulation of the skew term contributes to this and exploring alternative formulations for skew. For example, we could define a skew term using Mardia skewness (Mardia, 1970) or Kollo skewness (Kollo, 2008). Assuming we could formulate a (pseudo)metric using one of these measures of skewness, it would be interesting to see how it changes the performance of SID.

**Exploring other methods for dimensionality reduction**
We could also use random, isotropic, Gaussian projections to reduce the dimensionality of our features. To do this, we would simply create a matrix $U_k \in \mathbb{R}^{k \times d}$, which takes the $d$-dimensional data and reduces it to $k$-dimensional data, by sampling from an isotropic Gaussian. This matrix is computationally inexpensive to construct and has several nice statistical properties. For example, $U_k$ is likely to be a projection matrix and unitary if $k$ is small enough (Wegner, 2021).

**Identify other applications**
GAN evaluation is not the only area where people use features extracted from a backbone architecture and assume that these features are normally distributed. There is also few-shot learning (see Hu et al. (2021) and Chobola et al. (2021)) and out-of-distribution detection (see Lee et al. (2018) and Ahuja et al. (2019)). Both the potential speed-up of applying PCA to the features and computing skewness could be of interest in these domains.

## Broader Impacts

Generative models are powerful tools which offer both positive applications and opportunities for misuse. While this paper focused on the mathematical aspect of generative model evaluation, we recognize the importance of considering potential societal impacts of our work.

Evaluation measures like SID can be used to improve generative models, some of which might be used for nefarious purposes, such as generating fake social media profiles to deceive users. Additionally, all measures have their shortcomings, and there may be hidden biases in SID which have yet to be identified. We have not yet explored, for instance, how SID handles biases in training data. These are important considerations not only for this work but for the broader field of generative model evaluation.

## Acknowledgments

RM was partially supported by the NSF grant DMS-2307971, and by a grant from the Simons Foundation MP-TSM-00002904.

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

## A   Related work

The landscape of GAN evaluation measures has seen substantial growth since FID was introduced in 2017. This discussion focuses on evaluation metrics most closely related to ours; for a comprehensive overview, see Borji (2022).

There are relatively few research works that address the assumption that the Inception-v3 features are Gaussian. Swisher (2022) reduces the dimensionality of the features using an autoencoder to be able to directly compute the optimal transport distance between the distributions in the latent space. This issue is also addressed in (Tsitsulin et al., 2019), which rather than using CNN embeddings builds a $k$-nearest neighbors graphs directly from the data, and then estimates the differences between those graphs. There is also a class-aware version of FID (Liu et al., 2018) that tries to model different modes as Gaussians. Additionally, in (Luzi et al., 2023), the authors develop a new metric, WaM, to compare image features, using Gaussian mixture models (GMMs) instead. Since WaM uses GMMs, it is a generalization of FID.

Our work also shows that PCA can be used to speed up the computation time of FID. The computation time of FID is slow due to both the term $\mathrm{Tr}(\sqrt{\Sigma_1 \Sigma_2})$ and computing the Inception-v3 embeddings (see Mathiasen & Hvilshøj (2020, Section 2)). There are several recent papers that address this issue. Mathiasen & Hvilshøj (2020) propose an algorithm to compute the term $\mathrm{Tr}(\Sigma_1 \Sigma_2)$ by constructing a matrix with the same eigenvalues. Motivated by this work and applications of random matrix theory to deep learning, Wu & Koelzer (2022) proposed comparing the sorted eigenvalues of the two covariance matrices. They show that this is much faster and appears to converge faster than FID, requiring a smaller amount of data to estimate accurately.

## B Proofs of homeomorphisms

*Proof of Example 1.* Let $\mathbb{P} = \{p_\lambda : \lambda \in (0, \infty)\}$ be the distribution space. Then, $f : (0, \infty) \to \mathbb{P}$ defined pointwise by $f(\lambda) = p_\lambda$ gives the desired forward mapping. Essentially, given $\lambda$, one can easily construct the exponential distribution $p_\lambda$ by the pointwise definition in Equation (3).

Now we construct the inverse of $f$. Define $g : (0, \infty) \to \mathbb{P}$ by $g(p_\lambda) = p_\lambda(0) = \lambda$. We see that

$$(f \circ g)(p_\lambda) = f(g(p_\lambda)) = f(\lambda) = p_\lambda$$

and

$$(g \circ f)(\lambda) = g(f(\lambda)) = g(p_\lambda) = \lambda$$

implying that $f$ and $g$ are inverses of each other.

Thus, exponential distributions are completely characterized by their parameter $\lambda$ and consequently are completely characterized by their mean $\frac{1}{\lambda}$.

Now we show that the bijection is a homeomorphism. Let $\epsilon > 0$ be given along with $\lambda \in (0, \infty)$. Letting $\delta = -2\epsilon^2 + 2\epsilon\sqrt{\epsilon^2 + \lambda}$, we see that for any $\lambda' \in (0, \infty)$ such that $|\lambda - \lambda'| < \delta$ we have

$$\begin{aligned}
\|p_\lambda - p_{\lambda'}\|_{L^2(\mathbb{R}_{>0})} &= \sqrt{\int_0^\infty \left(\lambda\exp(-\lambda x) - \lambda'\exp(-\lambda'x)\right)^2 dx} \\
&= \sqrt{\int_0^\infty \left(\lambda^2\exp(-2\lambda x) - 2\lambda\lambda'\exp(-x(\lambda+\lambda')) + \lambda'^2\exp(-2\lambda'x)\right)dx} \\
&= \sqrt{\frac{\lambda}{2} - \frac{2\lambda\lambda'}{\lambda+\lambda'} + \frac{\lambda'}{2}} \\
&= \sqrt{\frac{\lambda^2 - 2\lambda\lambda' + \lambda'^2}{2(\lambda+\lambda')}} \\
&= \frac{|\lambda - \lambda'|}{\sqrt{2\lambda + 2\lambda'}} \\
&< \frac{\delta}{2\sqrt{\lambda - \delta}} \\
&= \frac{-2\epsilon^2 + 2\epsilon\sqrt{\epsilon^2 + \lambda}}{2\sqrt{\lambda + 2\epsilon^2 - 2\epsilon\sqrt{\epsilon^2 + \lambda}}} \\
&= \sqrt{\frac{(-\epsilon^2 + \epsilon\sqrt{\epsilon^2 + \lambda})^2}{\lambda + 2\epsilon^2 - 2\epsilon\sqrt{\epsilon^2 + \lambda}}} \\
&= \epsilon\sqrt{\frac{\epsilon^2 + \epsilon^2 + \lambda - 2\epsilon\sqrt{\epsilon^2 + \lambda}}{\lambda + 2\epsilon^2 - 2\epsilon\sqrt{\epsilon^2 + \lambda}}} \\
&= \epsilon.
\end{aligned}$$

Thus the mapping is continuous in one direction. Now, focusing on the inverse, first note that we must pick an $\delta > 0$ so that all $p_{\lambda'} \in \mathbb{P}$ satisfy

$$\|p_\lambda - p_{\lambda'}\|_{L^2(\mathbb{R}_{>0})} = \sqrt{\int_0^\infty \left(\lambda\exp(-\lambda x) - \lambda'\exp(-\lambda'x)\right)^2 dx} = \frac{|\lambda - \lambda'|}{\sqrt{2\lambda + 2\lambda'}} < \delta$$

We see that after we pick such a $\delta > 0$, we will have

$$|\lambda - \lambda'| < \delta\sqrt{2(\lambda+\lambda')} \leq 2\delta\sqrt{\lambda + \delta}$$

If we pick an appropriate $\delta > 0$, so that $2\delta\sqrt{\lambda + \delta} = \epsilon$, then our proof is done. To do this, we square both sides and try to solve the cubic equation:

$$\delta^3 + \lambda\delta^2 - \frac{\epsilon^2}{4} = 0.$$

However, we need not find the explicit formula for the correct $\delta > 0$. Although such a formula can be found, it is very messy and not very enlightening. We must just verify that such a $\delta > 0$ exists. Note that if we choose $\delta = 0$, then the left-hand side of the equation becomes negative. Hence, there must be a positive, real $\delta > 0$ that solves that equation (since it is a cubic polynomial). Call this $\delta^*$. Then, we have that

$$|\lambda - \lambda'| < 2\delta^*\sqrt{\lambda + \delta^*} = \epsilon,$$

as desired. $\qquad\square$

*Proof of Example 2.* Let $\mathbb{P} = \{p_{\boldsymbol{\mu},\boldsymbol{\Sigma}} : \boldsymbol{\mu} \in \mathbb{R}^p, \boldsymbol{\Sigma} \in \mathbf{S}_{++}^p\}$ be the distribution space. First, we define a function $m : \mathbb{P} \to \mathbb{R}^p$ which takes $p_{\boldsymbol{\mu},\boldsymbol{\Sigma}}$ and returns the mean $\boldsymbol{\mu}$. We can get the mean from $p_{\boldsymbol{\mu},\boldsymbol{\Sigma}}$ by selecting $\boldsymbol{x} \in \mathbb{R}^p$ which satisfies the following:

$$\sup_{\boldsymbol{x}\in\mathbb{R}^p} p_{\boldsymbol{\mu},\boldsymbol{\Sigma}}(\boldsymbol{x}).$$

We can see from Equation (4) that $\boldsymbol{\mu}$ always satisfies this supremum. To see uniqueness, note that

$$(\boldsymbol{x}_1 - \boldsymbol{\mu})^\top \boldsymbol{\Sigma}^{-1}(\boldsymbol{x}_1 - \boldsymbol{\mu}) = (\boldsymbol{x}_2 - \boldsymbol{\mu})^\top \boldsymbol{\Sigma}^{-1}(\boldsymbol{x}_2 - \boldsymbol{\mu}) = 0$$

implies that $\boldsymbol{x}_1 = \boldsymbol{x}_2 = \boldsymbol{\mu}$ since $\boldsymbol{\Sigma}$ is positive definite. So we define a function $m : \mathbb{P} \to \mathbb{R}^p$ by $m(p_{\boldsymbol{\mu},\boldsymbol{\Sigma}}) = \arg\max_{\boldsymbol{x}\in\mathbb{R}^p} p_{\boldsymbol{\mu},\boldsymbol{\Sigma}}(\boldsymbol{x})$.

Next, we construct a function $s : \mathbb{P} \to \mathbf{S}_{++}^p$ that takes $p_{\boldsymbol{\mu},\boldsymbol{\Sigma}}$ and returns the covariance $\boldsymbol{\Sigma}$. Note that due to our construction of $m$, we can easily determine the value of $|2\pi\boldsymbol{\Sigma}|^{-\frac{1}{2}} = p_{\boldsymbol{\mu},\boldsymbol{\Sigma}}(m(p_{\boldsymbol{\mu},\boldsymbol{\Sigma}}))$. Therefore, we focus on the simplified function

$$p'_{\boldsymbol{\mu},\boldsymbol{\Sigma}}(\boldsymbol{x}) = \boldsymbol{x}^\top \boldsymbol{\Sigma}^{-1}\boldsymbol{x} = \sum_{i=1}^p \sum_{j=1}^p x_i x_j \Sigma_{ij}^{-1} \tag{8}$$

which can be tied back to $p_{\boldsymbol{\mu},\boldsymbol{\Sigma}}$ via the relation $p'_{\boldsymbol{\mu},\boldsymbol{\Sigma}}(\boldsymbol{x}) = -2\log\left(|2\pi\boldsymbol{\Sigma}|^{\frac{1}{2}} p_{\boldsymbol{\mu},\boldsymbol{\Sigma}}(\boldsymbol{x} + \boldsymbol{\mu})\right)$. In the above expression, $\Sigma_{ij}^{-1}$ is the $ij$-th entry in $\boldsymbol{\Sigma}^{-1}$. Therefore, we first construct the simplified version of $s$ and call it $s' : p'_{\boldsymbol{\mu},\boldsymbol{\Sigma}} \mapsto \boldsymbol{\Sigma}$ as follows.

We can simply obtain the diagonal of $\boldsymbol{\Sigma}^{-1}$ from $p'_{\boldsymbol{\mu},\boldsymbol{\Sigma}}$. Let $\boldsymbol{x}_i$ be the zero vector except that the $i$-th element is 1. Then, we see from Equation (8) that $p'_{\boldsymbol{\mu},\boldsymbol{\Sigma}}(\boldsymbol{x}_i) = \Sigma_{ii}^{-1}$. Similarly, let $\boldsymbol{x}_{ij}$ be the zero vector, except that the elements $i$-th and $j$-th are 1. Then, we see from Equation (8) that $p'_{\boldsymbol{\mu},\boldsymbol{\Sigma}}(\boldsymbol{x}_{ij}) = \Sigma_{ii}^{-1} + 2\Sigma_{ij}^{-1} + \Sigma_{jj}^{-1}$. We rearrange this expression to get $\Sigma_{ij}^{-1} = \frac{1}{2}\left(p'_{\boldsymbol{\mu},\boldsymbol{\Sigma}}(\boldsymbol{x}_{ij}) - \Sigma_{ii}^{-1} - \Sigma_{jj}^{-1}\right)$. By doing this for each $i, j \in \{1, \ldots, p\}$, we get $\boldsymbol{\Sigma}^{-1}$ and thus $\boldsymbol{\Sigma}$.

We construct $s$ as follows. First, we start $p_{\boldsymbol{\mu},\boldsymbol{\Sigma}}$ and calculate $\boldsymbol{\mu}$ from $m$. Second, we use $p_{\boldsymbol{\mu},\boldsymbol{\Sigma}}$ and $\boldsymbol{\mu}$ to construct $p'_{\boldsymbol{\mu},\boldsymbol{\Sigma}}$. Lastly, we use $p'$ as described above to calculate $\boldsymbol{\Sigma}$ as desired.

We have constructed functions which, when used together, map from $\mathbb{P}$ to $\mathbb{R}^p \times \mathbf{S}_{++}^p$ as desired. $\qquad\square$

## C  Invertible transformations applied to the Frobenius metric

We first recall the Theorem from 2.2 which shows that the third term in Equation (6), $\| \sqrt[\circ 3]{\mathbf{S}_1} - \sqrt[\circ 3]{\mathbf{S}_2}\|_F^2$, is a squared metric.

**Theorem 1 .** *For* $\mathbf{A}, \mathbf{B} \in \mathbb{X}(n_1, \ldots, n_r; \mathbb{C})$ *we have that*

$$d(\mathbf{A}, \mathbf{B}) := \sqrt{\sum_{\boldsymbol{i}\in\{1,\ldots,n_1\}\times\ldots\times\{1,\ldots,n_r\}} \left|\alpha(\mathbf{A}_{\boldsymbol{i}}) - \alpha(\mathbf{B}_{\boldsymbol{i}})\right|^2}$$

*defines a proper metric* $d$.

*Proof of Theorem 1.* We first show that $d$ is non-negative, symmetric, and that $d(\mathbf{A}, \mathbf{B}) = 0$ if and only if $\mathbf{A} = \mathbf{B}$. Clearly, $d(\mathbf{A}, \mathbf{B}) \geq 0$ for all $\mathbf{A}, \mathbf{B}$ because the sum is over non-negative values. The symmetry is also clear from the definition because $\left|\alpha(\mathbf{A_i}) - \alpha(\mathbf{B_i})\right|^2 = \left|\alpha(\mathbf{B_i}) - \alpha(\mathbf{A_i})\right|^2$ for each index $i$. If $\mathbf{A} = \mathbf{B}$ then $d(\mathbf{A}, \mathbf{B}) = 0$. Conversely, if $d(\mathbf{A}, \mathbf{B}) = 0$, this implies that each term in the sum $\left|\alpha(\mathbf{A_i}) - \alpha(\mathbf{B_i})\right|^2 = 0$, which implies that $\mathbf{A} = \mathbf{B}$ because $\alpha$ is a bijection, as desired.

All that is left to show is that $d$ satisfies the triangle inequality. So we introduce a new array $\mathbf{C} \in \mathbb{X}(n_1, \ldots, n_r; \mathbb{C})$ for the proof. We write the vector $\boldsymbol{a} \in \mathbb{C}^{n_1 \times \cdots \times n_r}$ (and its elements $a_j$) as vectorized (or flattened) versions of $\mathbf{A}$. Similar definitions apply to $\mathbf{B}$ and $\mathbf{C}$.

$$
\begin{aligned}
d(\mathbf{A}, \mathbf{B}) + d(\mathbf{B}, \mathbf{C}) &= \sqrt{\sum_i \left|\alpha(\mathbf{A_i}) - \alpha(\mathbf{B_i})\right|^2} + \sqrt{\sum_i \left|\alpha(\mathbf{B_i}) - \alpha(\mathbf{C_i})\right|^2} \\
&= \sqrt{\sum_j \left|\alpha(a_j) - \alpha(b_j)\right|^2} + \sqrt{\sum_j \left|\alpha(b_j) - \alpha(c_j)\right|^2} \\
&= \|\alpha(\boldsymbol{a}) - \alpha(\boldsymbol{b})\|_2 + \|\alpha(\boldsymbol{b}) - \alpha(\boldsymbol{c})\|_2 \\
&\geq \|\alpha(\boldsymbol{a}) - \alpha(\boldsymbol{c})\|_2 \\
&= \sqrt{\sum_j \left|\alpha(a_j) - \alpha(c_j)\right|^2} \\
&= \sqrt{\sum_i \left|\alpha(\mathbf{A_i}) - \alpha(\mathbf{C_i})\right|^2} \\
&= d(\mathbf{A}, \mathbf{C}),
\end{aligned}
$$

where we denote $\alpha(\boldsymbol{a}), \alpha(\boldsymbol{b})$, and $\alpha(\boldsymbol{c})$ as $\alpha$ applied element-wise to the vectors $\boldsymbol{a}, \boldsymbol{b}$, and $\boldsymbol{c}$. Moreover, the inequality comes from the 2-norm on vectors here. Thus, we have shown that by flattening these multi-dimensional arrays, we can use properties of vector metrics to prove the triangle inequality. □

## D Discussion of normalization

PCA can be computed by first computing the singular value decomposition (SVD) of the centered data matrix. Suppose that $n > p$ and that the data matrix $\boldsymbol{X} \in \mathbb{R}^{n \times p}$ is centered. Then its covariance matrix is $\widehat{\boldsymbol{\Sigma}} = \frac{1}{n-1}\boldsymbol{X}^\top \boldsymbol{X}$. If the singular value decomposition yields $\boldsymbol{X} = \boldsymbol{U}\begin{bmatrix}\boldsymbol{D}\\\boldsymbol{0}\end{bmatrix}\boldsymbol{V}^\top$ with $\boldsymbol{U} \in \mathbb{R}^{n \times n}, \boldsymbol{D} \in \mathbb{R}^{p \times p}$, and $\boldsymbol{V} \in \mathbb{R}^{p \times p}$ then

$$
\widehat{\boldsymbol{\Sigma}} = \frac{1}{n-1}\boldsymbol{V}\boldsymbol{D}^2\boldsymbol{V}^\top. \tag{9}
$$

To perform PCA, we split up $\boldsymbol{V} = \begin{bmatrix}\boldsymbol{V}_1 & \boldsymbol{V}_2\end{bmatrix}$ and right multiply $\boldsymbol{X}$ by $\boldsymbol{V}_1 \in \mathbb{R}^{p \times k}$. Here, the number of columns $k$ of $\boldsymbol{V}_1$ is the dimension we are reducing to; equivalently, it is the number of principal components.

So, given two matrices of features $\boldsymbol{X}$ and $\boldsymbol{Y}$, we compute the singular value decomposition of the reference data set $\boldsymbol{Z}$. $\boldsymbol{Z}$ may be a concatenation of $\boldsymbol{X}$ and $\boldsymbol{Y}$, or it may be larger. Writing $\boldsymbol{Z} = \boldsymbol{U}\begin{bmatrix}\boldsymbol{D}\\\boldsymbol{0}\end{bmatrix}\boldsymbol{V}^\top$, we reduce $\boldsymbol{X}$ and $\boldsymbol{Y}$ by right multiplication by $\boldsymbol{V}_1$, the key point here being that we are using the same matrix $\boldsymbol{V}_1$ to reduce both $\boldsymbol{X}$ and $\boldsymbol{Y}$.

If we consider the covariance of $\boldsymbol{X}\boldsymbol{V}_1$ we get

$$
\widehat{\boldsymbol{\Sigma}}_1 = \frac{1}{n-1}\boldsymbol{V}_1^\top \boldsymbol{X}^\top \boldsymbol{X}\boldsymbol{V}_1 = \frac{1}{n-1}\boldsymbol{V}_1^\top \left(\boldsymbol{V}_1\boldsymbol{D}_1\boldsymbol{V}_1^\top + \boldsymbol{V}_2\boldsymbol{D}_2\boldsymbol{V}_2^\top\right)\boldsymbol{V}_1 = \frac{1}{n-1}\boldsymbol{D}_1
$$

since $\boldsymbol{V}_1^\top \boldsymbol{V}_1 = \boldsymbol{I}_k$ and $\boldsymbol{V}_2^\top \boldsymbol{V}_1 = \boldsymbol{0}$. Note that

$$
\operatorname{Tr}\widehat{\boldsymbol{\Sigma}} = \frac{1}{n-1}\sum_{i=1}^{p}[\boldsymbol{D}]_{ii}, \text{ but } \operatorname{Tr}\widehat{\boldsymbol{\Sigma}}_1 = \frac{1}{n-1}\sum_{i=1}^{k}[\boldsymbol{D}_1]_{kk},
$$

implying that the scales of the two traces has now changed.

We conclude that to normalize between calculations of FID with different values of $k$, one should use the following transformation of $\boldsymbol{X}$:

$$\boldsymbol{Y} = \sqrt{\frac{\sum_{i=1}^{p}[\boldsymbol{D}]_{ii}}{\sum_{i=1}^{k}[\boldsymbol{D}_1]_{kk}}} \boldsymbol{X}\boldsymbol{V}_1.$$

This results in the following covariance:

$$\frac{1}{n-1}\boldsymbol{Y}^\top\boldsymbol{Y} = \frac{1}{n-1}\frac{\sum_{i=1}^{p}[\boldsymbol{D}]_{ii}}{\sum_{i=1}^{k}[\boldsymbol{D}_1]_{kk}}\boldsymbol{D}_1$$

so that the trace of that quantity and of $\widehat{\boldsymbol{\Sigma}}$ are both $\frac{1}{n-1}\sum_{i=1}^{p}[\boldsymbol{D}]_{ii}$.

# E   Additional skew experiments

Here we show some additional experiments regarding skew; i.e., the blur experiment (Figure 3) and PCA dimensionality reduction on SID (Figure 4).

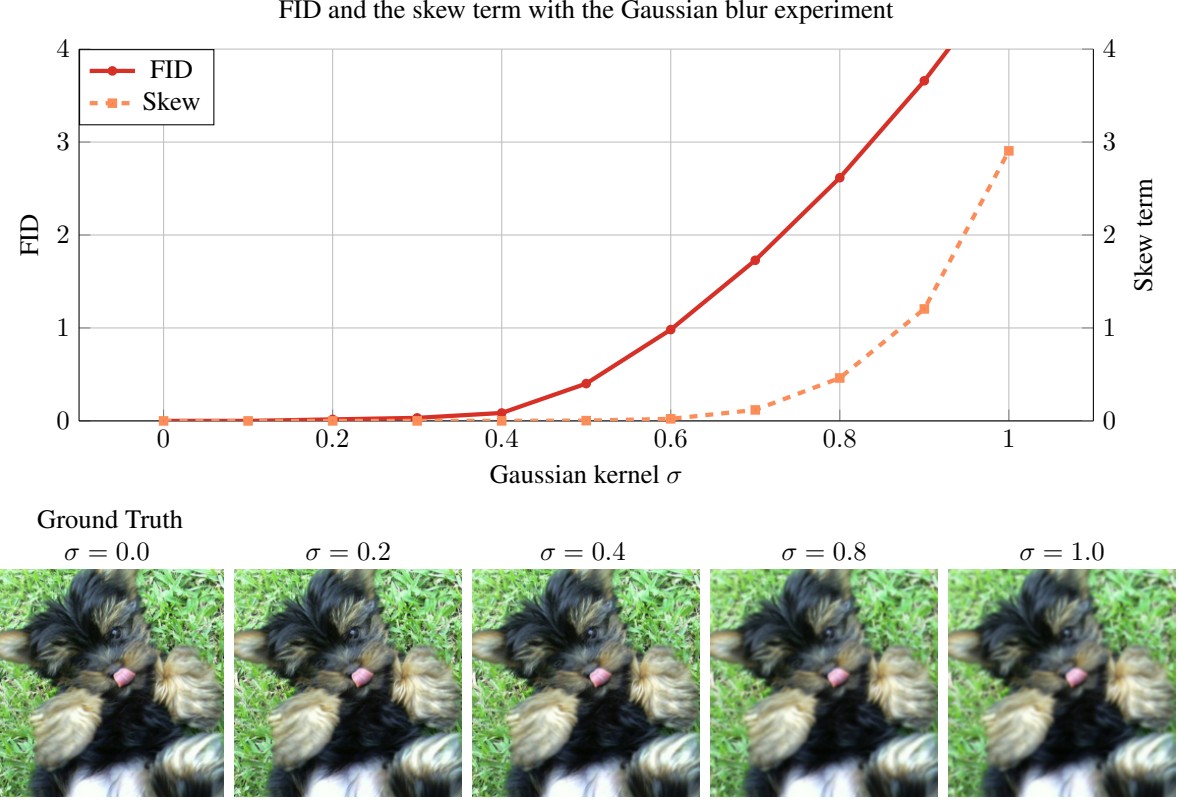

**Figure 3:** Skew tracks more with human perception than FID. Note that the blur levels on the images are quite low and undetectable until $\sigma > 0.4$ and skew stays low for these undetectable noise levels, in contrast to FID. These were typical images taken from the 50,000 samples used to calculate the terms.

# F   Using different models for feature extraction

In this section we investigate how using different architectures to extract features from data affects SID. In Figure 5 we plot the behavior of SID on four noise distortion experiments using Inception-v3, ResNet-18, ResNet-50,

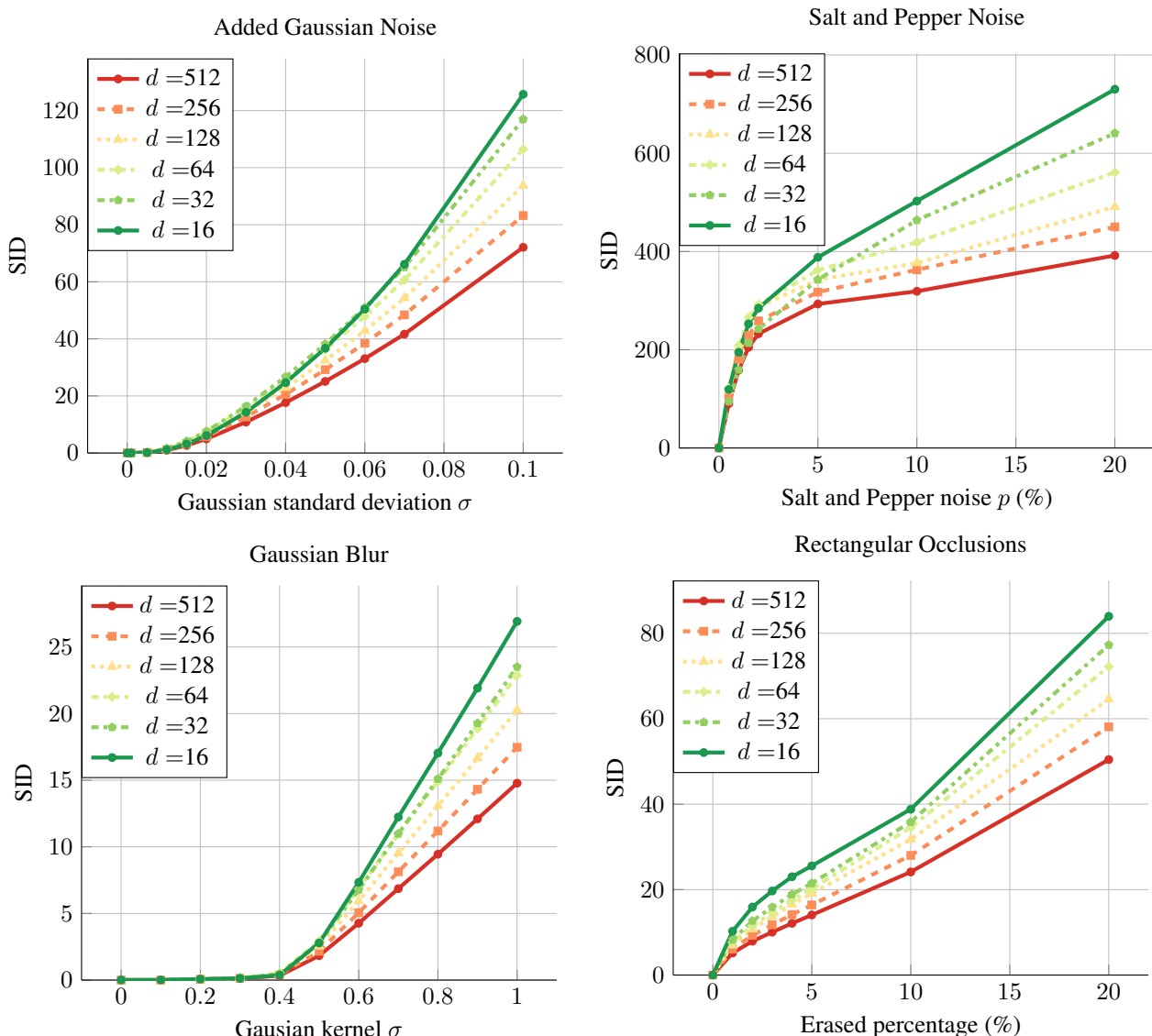

**Figure 4:** SID behaves similarly even when we project features down to lower dimensions via PCA. We show four types of distortions that are applied to ImageNet images: added Gaussian noise, salt and pepper noise, Gaussian blur, and rectangular occlusions. For the Gaussian blur, we use a kernel size of 5. In each plot, SID is shown as a function of the parameter controlling the distortion, after reducing the embeddings to dimension $d \in \{512, 256, 128, 64, 32, 16\}$. All experiments are done using Inception-v3 as a feature extractor on the entire (50K) ImageNet validation set.

and ResNeXt-101 (32×8d) as feature extractors. The noise distortions we use are additive Gaussian noise, salt and pepper noise, Gaussian blur, and rectangular occlusions, as described in Section 4.2. All the features come from ImageNet as in Figures 1 and 4. Additionally, dimensionality reduction is applied across all models to dimensions $d \in \{512, 256, 128, 64, 32, 16\}$, although its worth noting that ResNet-18 features are already 512 dimensional. We see that the curve shapes are relatively similar across different settings, with some variations in the salt and pepper noise experiment. Overall, the plots show that SID is relatively robust with respect to dimensionality reduction and choice of feature extraction model.

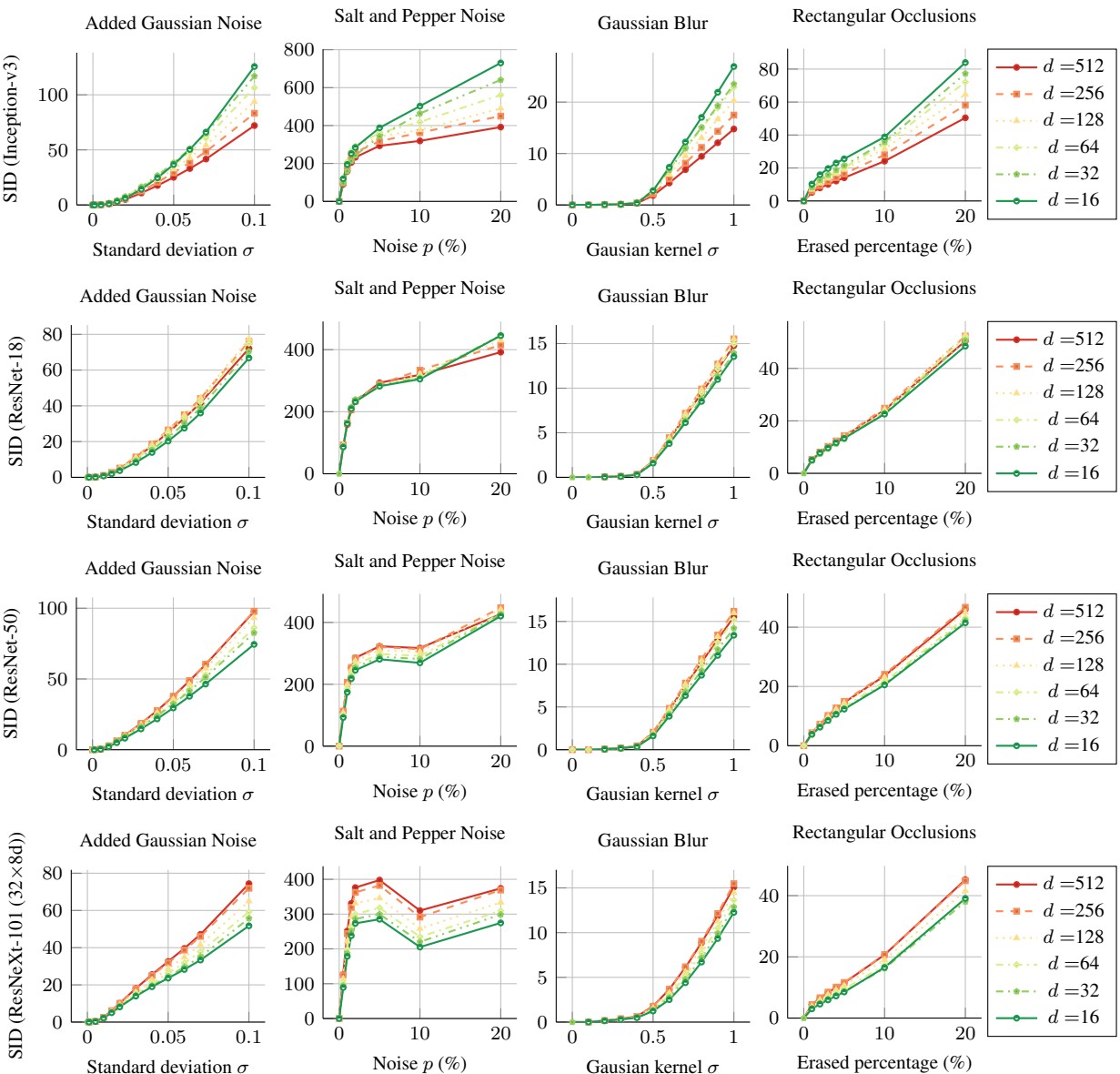

**Figure 5:** SID behaves similarly on noise distortion experiments across different choices of feature extractors. We show four types of distortion applied to ImageNet images, using Inception-v3, ResNet-18, ResNet-50, and ResNeXt-101 (32×8d) as feature extractors.

# G   Using SID on different datasets

We also study how the behavior of SID on noise distortion experiments is affected by changing the dataset. In Figure 5 we plot SID for the same four noise distortions (additive Gaussian noise, salt and pepper noise, Gaussian blur, and rectangular occlusions) on ImageNet, FFHQ, CelebA, and SVHN. All the features come from Inception-v3. As in our other experiments, we apply dimensionality reduction to dimensions $d \in \{512, 256, 128, 64, 32, 16\}$ for all the datasets. We see that all settings yield curves which increase as the noise distortion is increased, as desired.

We additionally perform the above experiments on data from diffusion models. The results are shown in Figure 7. We used pretrained diffusion models[4] that generated data from the Stanford Cars dataset Krause et al. (2013), CelebA Liu et al. (2015), AFHQ Choi et al. (2020), and the Flowers dataset Tung (2020).

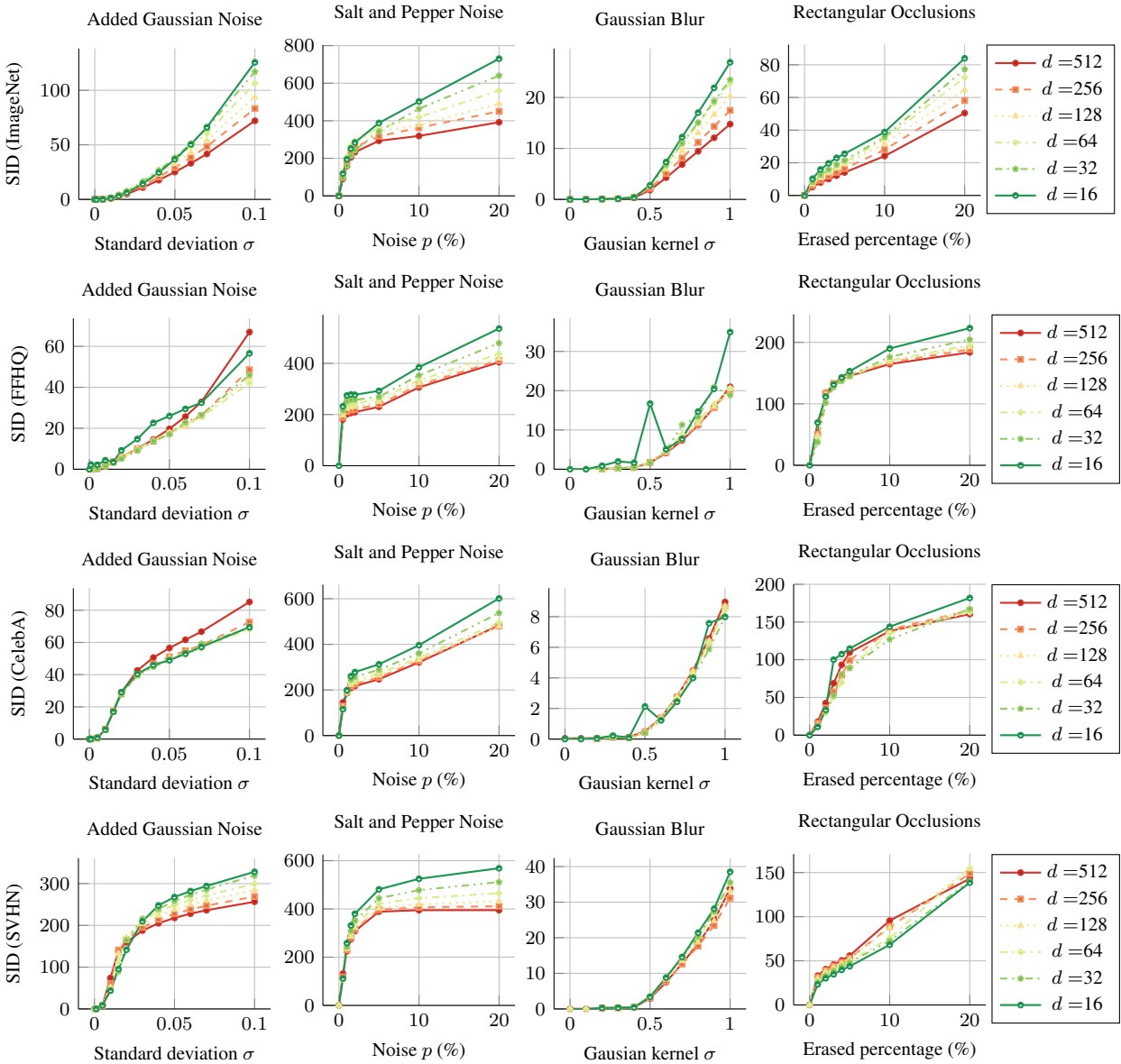

**Figure 6:** SID increases as the noise distortion is increased, irrespective of the choice of dataset. We show results from ImageNet, FFHQ, CelebA, and SVHN, using the Inception-v3 feature extractor.

---

[4]The pretrained diffusion models are from the repository: `https://github.com/VSehwag/minimal-diffusion`

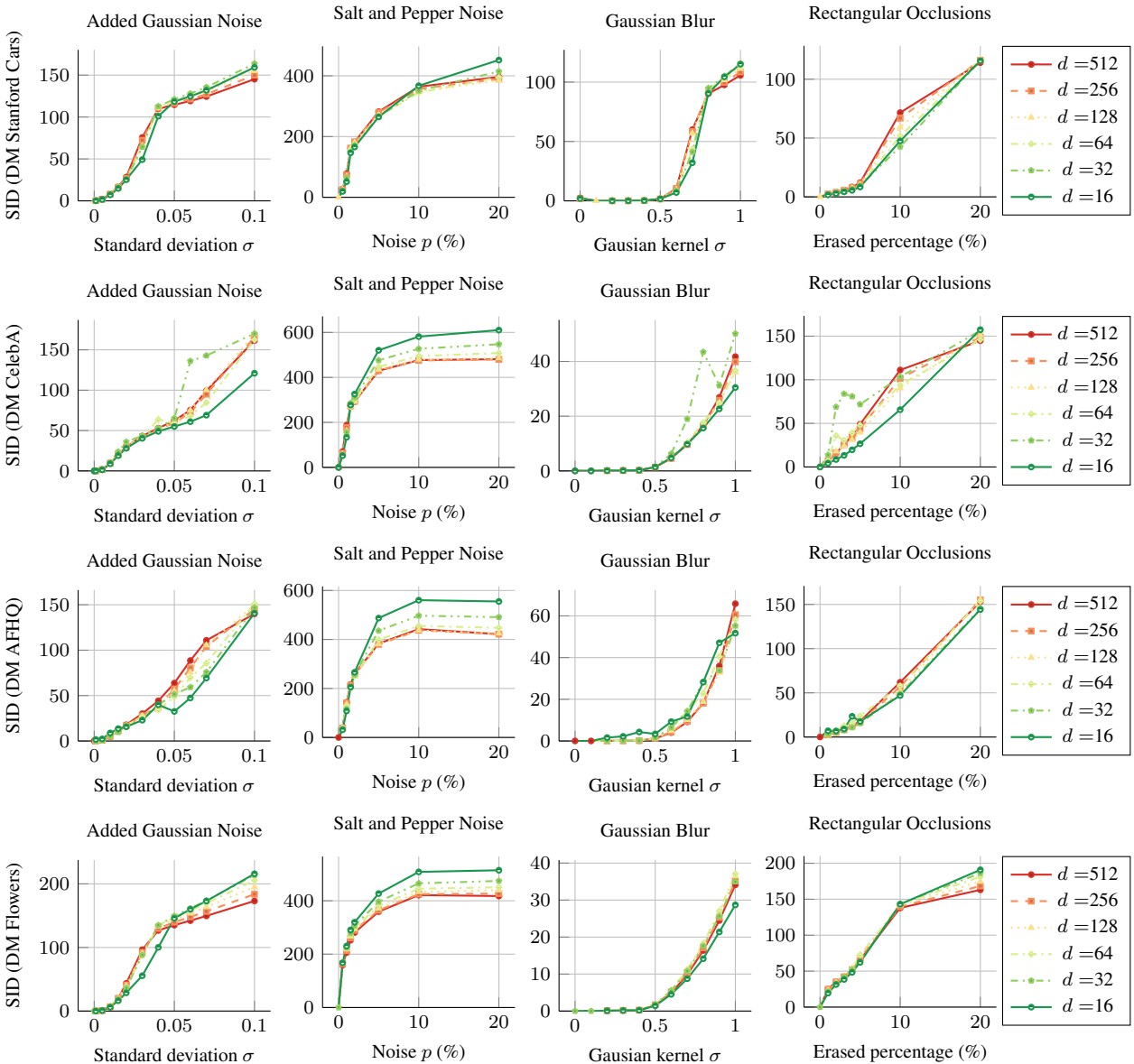

**Figure 7:** SID behaves as expected with different datasets; increasing as we increase the noise distortions. Here we use data from four pretrained diffusion models trained on the Stanford Cars dataset, CelebA, AFHQ, and the Flowers dataset. All features were obtained with the Inception-v3 feature extractor.

