# OpenReview forum: "Using Skew to Assess the Quality of GAN-generated Image Features"
_TMLR — Accepted by TMLR_

### Review · Reviewer_vu1z · 2024-01-04

**Summary Of Contributions:**

In this paper, the authors develop a new metric, named Skew Inception Distance (SID), for the image generation evaluation. Compared with the Fréchet Inception Distance (FID), the SID metric can account for the importance of third-moments in image features for the skewed Inception-v3 features. Some evaluations are performed to show the experimental advantages over FID.

**Audience:**

Yes

**Broader Impact Concerns:**

Although I don't observe any major concerns regarding impact, it would enhance the paper if the authors incorporate brief discussions on the social implications.

**Claims And Evidence:**

No

**Requested Changes:**

Please see the weakness part.

**Strengths And Weaknesses:**

**Strength**:

+ The paper is well written. I can follow it easily.
+ Some theoretical analyses are provided.
+ The proposed SID metric has clear definitions and the authors also detailedly introduce the reason to use the PCA and its impacts on final results.

**Weakness**:

+ The proposed SID looks to be an incremental extension of FID.
+ This submission only compares with FID. So could the authors also make some comparisons with other popular metrics? This can better present the empirical advantages of the proposed metric.
+ The submission needs to measure the alignment with human perception. But in Figure 2, I am a bit confused about why to conclude "Skew tracks more with human perception than FID". Could the authors provide more details? And a user study might help here for a solid conclusion.
+ I am also worried about the experiment part. It seems that the authors only perform the comparisons with the Inception-v3 features. How about the performance with other backbone networks like ResNet-50 or ViT? Since SID is a straightforward extension of FID, it is almost necessary to conduct comprehensive experiments to more convincingly demonstrate the empirical advantages of SID.
+ The authors only apply the corrupted images to measure the differences between SID and FID. I wonder about the quantitative differences for really generated images. For example, the authors can use GAN or diffusion models to generate 1000 images and then compute FID \& SID. It is important to indicate that SID is statistically meaningful by comparing over a large group of generation results across different benchmarks.
+ In Section 4.1, could the authors further elaborate how to conclude "features are skewed after dimensionality reduction using PCA"? I know some tests using Mardia skewness test and Kolmogorov–Smirnov hypothesis tests are conducted, could the authors provide more details to make papers more self-contained?
+ In the legend of Figure 1, I can guess that the **d** (e.g. d=16) indicates the dimension. But I think it would be better if its meaning can be clearly specified in the figure caption.
+ Maybe the paper could be improved by adding a limitation section. Some discussions about the failure cases are also welcome.

---

> ### Author Response · Authors · 2024-02-22
> **Official response to reviewer vu1z**
>
> Thank you for your helpful feedback and suggestions. To start, we'll discuss the additional experiments we conducted, as this is the most substantial improvement we made to the paper in response to the feedback we received. Following your suggestion to conduct experiments with other backbone networks, we performed experiments where we calculate SID using features from ResNet-18, ResNet-50, and ResNeXt-101 (32x8d) in addition to Inception-v3. We concluded that SID is robust to changing the feature extractor and dimensionality reduction (for all four networks, we reduce the dimensions to d = 512, 256, 128, 64, 32, and 16). We also evaluated the noise distortions discussed in our paper on FFHQ, CelebA, and SVHN. Finally, we used pretrained diffusion models that were trained on Stanford Cars, CelebA, AFHQ, and the Flowers dataset to generate 50,000 images. We then calculated SID on these images before and after image distortions, with the same dimension reduction. SID behaves as expected on all of these, with SID increasing as the noise distortion is increased. All results are in the Appendix and referenced in the main text at the end of Section 4.
>
> We'll address your other comments in the order they appear:
> - We agree that SID is incremental, and sincerely hope that our paper does not suggest otherwise. Please let us know if we make any unsubstantiated claims; we are committed to ensuring the accuracy and clarity of our work.
> - To clarify, in Figure 2 the assertion that the noise levels on the images are quite low and undetectable until $\sigma > 0.015$ is based on our judgement.
> - Concerning the suggestion to make comparisons with other popular metrics, we decided this would not align well with the focus of our paper. Our paper aims to introduce SID to address an important limitation of FID, and we do not make claims about how SID compares to other metrics. We agree this is an interesting and important direction for future work, and added the following to the future works section: “The current work does not compare SID to other established metrics, which is an important direction for future study.”
> - We clarified the way in which we conclude that the features are skewed by adding a footnote giving a high-level description of the Kolmogorov-Smirnov hypothesis test: "The Kolmogorov-Smirnov hypothesis test is a nonparametric method for evaluating the goodness-of-fit. It assesses whether an underlying probability distribution (the marginal distributions in our case) differs from a hypothesized distribution, a Gaussian distribution."
> - In response to your comment about the legend of Figure 1, we edited the figure caption in Figures 1 and 3 to include, "In each plot, FID is shown as a function of the parameter controlling the distortion, after reducing the embeddings to dimension $d \in {512, 256, 128, 64, 32, 16}$."
> - The limitations are addressed in the first paragraph of the conclusion. In our original submission we discussed two limitations. In our revision we add: ``A notable limitation of SID is its computational complexity, which is $O(d^3)$, where $d$ is the dimensionality of the features, making dimensionality reduction essential for its calculation. Consequently, our estimation of SID may not fully capture the original features.''
> - We added a Broader Impacts section to the end of our paper.

---

### Review · Reviewer_MLqA · 2024-01-05

**Summary Of Contributions:**

The paper introduces the Skew Inception Distance (SID), a novel metric for assessing GAN-generated image quality. SID advances beyond the Fréchet Inception Distance (FID) by accounting for non-Gaussian feature distributions, incorporating skewness into evaluation. Key contributions include:

1. **Theoretical Foundation**: Establishes SID as a pseudometric on probability distributions, providing a robust mathematical basis for its application.
2. **Improved Accuracy**: Demonstrates through experiments that SID aligns more closely with human perception than FID in certain scenarios.
3. **Efficiency Enhancement**: Introduces the use of PCA to accelerate computations for both FID and SID.
4. **Broader Applicability**: Suggests potential for SID's use in areas beyond GAN evaluation.

This work contributes to the field by offering a more nuanced and potentially accurate tool for evaluating the realism and quality of GAN-generated images.

**Audience:**

Yes

**Broader Impact Concerns:**

**Broader Impact Concerns:**

While the paper focuses on a technical advancement in evaluating GANs, it's essential to address potential ethical implications in a broader impact statement. This includes:

1. **Misuse of GANs**: Enhanced evaluation tools like SID could inadvertently aid in creating more realistic deepfakes or misleading content.
2. **Bias in Generated Images**: The paper should discuss how SID might handle biases inherent in training data, which can lead to unfair or biased outputs.

Including a thorough discussion of these aspects would ensure a responsible approach to the technology's broader societal impacts.

**Claims And Evidence:**

Yes

**Requested Changes:**

**Requested Changes:**

1. **Broaden Experimental Validation**: Include additional datasets to enhance SID's empirical validation. [Critical]
2. **Clarify Implementation**: Provide detailed implementation guidelines and computational cost comparison with FID. [Critical]
3. **Extended Comparative Analysis**: Compare SID with other metrics beyond FID for a more comprehensive evaluation. [Strengthening]
4. **Real-World Applications**: Explore and document potential applications of SID in real-world scenarios. [Strengthening]
5. **Robustness Testing**: Conduct robustness tests against various types of noise and image distortions. [Strengthening]
6. **Inclusion of Diffusion Model Experiments**: Conduct experiments using diffusion models to evaluate SID's performance in a broader range of generative models. This would not only strengthen the paper but also provide insights into SID's applicability across different types of generative approaches. [Critical]

These changes are aimed at enhancing the paper's rigor, relevance, and practical applicability.

**Strengths And Weaknesses:**

**Strengths:**

1. . **Theoretical Rigor**: Solid mathematical foundation
2. **Empirical Validation**: Effective demonstration of SID's advantages over FID through experiments.

**Weaknesses:**

1. **Limited Scope of Experiments**: More diverse datasets and more experiments on new generative models such as diffusion models could further validate SID's effectiveness.
2. **Complexity and Practicality**: Implementation complexity and computational cost compared to FID needs clarification.
3. **Comparative Analysis**: A more comprehensive comparison with other metrics could strengthen the argument.

---

> ### Author Response · Authors · 2024-02-22
> **Official response to reviewer MLqA**
>
> Thank you for your helpful feedback and suggested changes. First, we agreed with the critique that our submission could benefit from more diverse datasets and experiments. To address this, we evaluated the noise distortions discussed in our paper on FFHQ, CelebA, and SVHN. Additionally, we used pretrained diffusion models that were trained on Stanford Cars, CelebA, AFHQ, and the Flowers dataset to generate images. In the new experiments, SID behaved as expected, with SID increasing as the noise distortion is increased. We added the results to the Appendix and reference them at the end of Section 4.
>
> Second, we clarified the computational cost of SID as compared to FID. We added the following paragraph to Section 3:
> "In addition, computing the coskewness tensor is the slowest of the three terms; the operation scales as $O(d^3)$, where $d$ is the dimensionality of the reduced feature space. In comparison, FID can be computed in $O(d^2m + m^3)$ time, where $m << d$. (Although this requires observing that it suffices to compute the eigenvalues of $\Sigma_1\Sigma_2$; many FID implementations compute
> Tr$\(\sqrt{ \Sigma_1 \Sigma_2} )$ using the scipy.linalg.sqrtm function, which requires computing the Schur decomposition of $\Sigma_1 \Sigma_2$ and takes $O(d^3)$ time). Because of the memory and computational obstacles of SID, we reduce the dimension of the Inception-v3 embeddings.''
> We also clarified the implementation at the end of section 3. Please let us know if anything remains unclear.
>
> Third, regarding the suggestion to make comparisons with other popular metrics, we decided this would not align well with the focus of our paper. Our paper aims to introduce SID to address an important limitation of FID, and we do not make claims about how SID compares to other metrics. We agree this is an interesting and important direction for future work, and added the following to the future works section: “The current work does not compare SID to other established metrics, which is an important direction for future study.”
> Similarly, we considered exploring real world applications as you suggested. Although that is an interesting research direction, we feel that it is out of the scope of our paper. The last paragraph in our paper hints to some possible real world applications which would be great research directions, but would be entire papers by themselves.
>
> To address your concern about robustness testing, we added an experiment where we calculate SID using features from ResNet-18, ResNet-50, and ResNeXt-101 (32x8d) in addition to Inception-v3. We conclude that SID is robust to changing the feature extractor and dimensionality reduction (for all four networks, we reduce the dimensions to d = 512, 256, 128, 64, 32, and 16). More details are given in Appendix F.
>
> Finally, we appreciated your thoughts about the broader impacts, and added a Broader Impacts section to the end of our paper:
> ``Generative models are powerful tools which offer both positive applications and opportunities for misuse. While this paper focused on the mathematical aspect of generative model evaluation, we recognize the importance of considering potential societal impacts of our work.
>
> Evaluation measures like SID can be used to improve generative models, some of which might be used for nefarious purposes, such as generating fake social media profiles to deceive users. Additionally, all measures have their shortcomings, and there may be hidden biases in SID which have yet to be identified. We have not yet explored, for instance, how SID handles biases in training data. These are important considerations not only for this work but for the broader field of generative model evaluation.”
>
> Thank you for your careful and well-organized feedback and we hope that our changes are satisfactory.

---

### Review · Reviewer_A5ec · 2024-02-07

**Summary Of Contributions:**

The authors studied the limitations of existing FID metric used for evaluating the GANs, and purpose toe use higher-order moments to access the quality of GAN-generated image features. The authors give a few examples on how existing FID metrics could fail, and they propose a revision to existing FID. The authors further consider incorporating PCA to the metric calculation to accelerate the computation. Evaluation shows that the proposed method (when combined with dimensionality reduction) is computationally and memory efficient, mostly aligns with FID, and is better in terms for human perception.

**Audience:**

Yes

**Claims And Evidence:**

No

**Requested Changes:**

On the dimensionality reduction effect:

The evaluation becomes computational and memory efficient when the dimension has been reduced from 2048 to 256. 10% drop in the accuracy can be a lot. I mean, considering the scenario that model A is slightly better than model B according SID (without dimensionality reduction). Is it possible that the merits diminished or even method B is better than method A after the dimensionality reduction?



I'm a bit confused on the difference between SID and SFD. Does SID = SFD applied with Inception network?



Evaluation is fairly limited. It should be tested for multiple models, on multiple datasets, and see how the evaluation agrees with existing metrics and publications.



The title is higher-order moments, but actually, the whole article is pretty constrained to be no more than the 3rd moment.

**Strengths And Weaknesses:**

The article is a combination theoretical justification and experiments, with a good starting point motivation. The examples provided by the authors are easily to follow to understand the limitations of the Fréchet Inception Distance (FID). The authors also pointed out a few action items and directions for future work.

So FID assesses ‘how similar is a group of images relative to another group of images’, and the authors claim that the default assumption on the Gaussian distribution is not realistic. Thus, higher order moments like skewness are needed.

After some experiments, the authors show that FID and SID are actually very much aligned, and the biggest difference is that the SID seems better matching the human perception than FID.

Does that mean the actually issue of FID is that when measuring group similarities, it does not take into account the human perception due to it does not using higher-order information? Or, do the authors intend to show using higher-order information can help (to some extent) evaluate/align with the human perception issue? More elaboration on this finding could help strengthen this article.


FID is a computational more efficient metric comparing to SID. So if human perception is the major benefit for using SID, why not stick to FID and incorporating other human perception metrics? In actual evaluation, we typically look at multiple metrics not only FID, and subject evaluation could also be included during evaluation.

---

> ### Author Response · Authors · 2024-02-22
> **Official response to reviewer A5ec**
>
> Thank you for your helpful feedback and suggested changes. First, you make a good point that by performing dimensionality reduction, we may distort the features significantly. Therefore, we have added the following limitation to our conclusion: “Consequently, the dimensionality reduction we perform in estimating SID may not include information that is not captured in the largest principal components.''
>
> You are correct that SID = SFD applied with the Inception network. We clarified the SID/SFD distinction by modifying the last sentence before Section 3 to say, "Although our pseudometric SFD is quite general, for the remainder of the paper, we will apply SFD to Inception-v3 embeddings and refer to this specific case as SID."
>
> In response to the critique that the evaluation is fairly limited, we added experiments that use different models for feature extraction and different datasets.
>
> *Other models*: Specifically, we performed experiments where we calculate SID using features from ResNet-18, ResNet-50, and ResNeXt-101 (32x8d) in addition to Inception-v3. We concluded that SID is robust to changing the feature extractor and dimensionality reduction (for all four networks, we reduce the dimensions to d = 512, 256, 128, 64, 32, and 16).
>
> *Other datasets*: We evaluated the noise distortions discussed in our paper on FFHQ, CelebA, and SVHN. Additionally, we used pretrained diffusion models that were trained on Stanford Cars, CelebA, AFHQ, and the Flowers dataset to generate images. In the new experiments, SID behaved as expected, with SID increasing as the noise distortion is increased.
>
> These results are in the Appendix and referenced at the end of Section 4.
>
>
> You have a good point about the title, and we changed it to “Using Skew to Assess the Quality of GAN-generated Image Features'' to better represent our work since we use skew and not all higher order moments.
>
> Finally in response to your question regarding FID, SID, and human perception, you are correct that we are intending to show that using higher-order information can help (to some extent) evaluate/align with the human perception issue. It’s true that one could use FID and incorporate other human perception metrics, but depending on the resources available, computing SID could be more feasible.
>
> Thank you for taking the time to read our paper and give thoughtful feedback. We hope that our revisions are satisfactory.

---

### Decision · Action_Editor_6XwE · 2024-03-18

**Recommendation:** Accept as is

**Comment:**

The paper proposes a new metric to measure the output quality of generative models. The proposed method - Skew Inception Distance (SID) - is meant to improve on FID by considering third moments (skew) in addition to the first two moments, better matching human perception.

Overall, the reviewer comments have mostly been addressed (including additional experiments) or flagged as limitations within the paper. As such, the reviewers are in agreement, leaning accept, as the claims in the paper are supported. The main concerns are with respect to the extent of novelty.

The final recommendation is to accept the paper to TMLR as is.

**Audience:**

The proposed paper should be of interest to some in TMLR's audience.

**Claims And Evidence:**

(Summary from reviewer MLqA) This paper provides a novel metric for assessing GAN-generated image quality. SID advances beyond the Fréchet Inception Distance (FID) by accounting for non-Gaussian feature distributions, incorporating skewness into evaluation. A new metric can improve the representation learning of generative models such as the recent representation learnt by diffusion models.

Overall, the reviewer comments have mostly been addressed or flagged as limitations within the paper. As such, the reviewers are in agreement, leaning accept, as the claims in the paper are supported. The main concerns are with respect to the extent of novelty.